# Lyophilic and Sorption Properties of Chitosan Aerogels Modified with Copolymers Based on Glycidyl Methacrylate and Alkyl Methacrylates

**DOI:** 10.3390/polym14132711

**Published:** 2022-07-01

**Authors:** Vitalia M. Yartseva, Olga A. Makevnina, Ekaterina B. Bryuzgina, Evgeny V. Bryuzgin, Viktor V. Klimov, Olga V. Kolyaganova, Dmitry E. Nikolitchev, Alexander V. Navrotsky, Ivan A. Novakov

**Affiliations:** 1Chemical Engineering Faculty, Volgograd State Technical University, Lenin Avenue, 28, 400005 Volgograd, Russia; yvita1995@gmail.com (V.M.Y.); makevnina98@inbox.ru (O.A.M.); cher-ekaterina18@yandex.ru (E.B.B.); vicklimov@gmail.com (V.V.K.); ollik86@mail.ru (O.V.K.); a-navrotskiy@yandex.ru (A.V.N.); president@vstu.ru (I.A.N.); 2Department of Chemistry, Lomonosov Moscow State University, Leninskiye Gory, 1, Building 3, 119991 Moscow, Russia; 3Department of Physics, N.I. Lobachevsky State University of Nizhny Novgorod, Gagarin Avenue, 23, 603950 Nizhny Novgorod, Russia; nikolitchev@gmail.com

**Keywords:** chitosan, aerogel, grafted copolymers, hydrophobicity, oleophilicity, sorption properties

## Abstract

This paper discusses the influence of the structure of copolymers based on glycidyl methacrylate and alkyl methacrylates with C_6_–C_18_ hydrocarbon side groups on the wettability and sorption properties of surface-modified chitosan aerogels. The grafting of copolymers onto the surface of aerogels was confirmed by elemental analysis, X-ray photoelectron spectroscopy, and Fourier-transform infrared spectroscopy. As a result of the modification, with an increase in the amount of the hydrocarbon substituent alkyl methacrylate, the surface of the resulting materials became hydrophobic with contact angles in the range of 146–157°. At the same time, the water absorption of the aerogels decreased by a factor of 30 compared to that for unmodified aerogels, while the sorption capacity for light oil, diesel fuel, and synthetic motor oil remained at the level of more than 30 g/g. Chitosan aerogels with grafted copolymers based on glycidyl methacrylate and alkyl methacrylates retain biodegradation capacity; however, compared to unmodified chitosan, this process has an induction period.

## 1. Introduction

Chitosan is a natural polysaccharide, a product of deacetylation of chitin contained in the shells of crustaceans and insects; it is a renewable, biocompatible and biodegradable polymer [1,2]. The environmental friendliness of this compound, combined with the film- and fiber-forming capacity, as well as solubility in aqueous acid media [3], make chitosan attractive for obtaining aerogels, i.e., materials with low density, high porosity, and high specific surface area [4,5,6,7,8]. The physical characteristics of porous materials in conjunction with the chitosan properties allow us to use these materials as sorbents for various contaminants including liquid hydrocarbons, oil and oil products.

The amino and hydroxy groups contained in the structure of chitosan make it hydrophilic [9] and limit its use as a petroleum and oil sorbent. However, this structural feature allows chemical modification of chitosan and materials based on it to control the lyophilic properties by performing reactions that are specific to the listed functional groups: N, O–alkylation [10], acylation [11,12], sulfatation [13], phosphorylation [14], quaternization [15], graft copolymerization, and others [16,17,18,19]. In particular, chitosan modification can be performed in solution before the material-forming stage. When implementing such an approach, the solubility of the resulting chitosan derivatives may be altered [20] and film-forming capacity may be decreased [21]. In addition, bulk modification considerably affects the mechanical, optical, and barrier properties of the resulting materials [22]. Another modification approach is the grafting of modifiers onto the surface of the preformed chitosan material, which generally does not result in any changes in the material properties [23]. We have previously investigated the surface modification of chitosan films by low molecular weight aldehydes. However, low-molecular-weight agents with small molecular sizes can penetrate deep into the films and affect not only the surface properties of the material, but also produce changes in the supramolecular structure [24].

Chemical modification of the chitosan surface by grafting synthetic polymers allows us to combine the characteristics of natural and synthetic macromolecules (e.g., adjusting lyophilic characteristics and imparting selective sorption properties) while maintaining the biodegradability of the polymer matrix. Synthetic macromolecules can be grafted onto the chitosan chain by radical polymerization with chemical or physical initiation [23,25], ring-opening polymerization [26], and cationic polymerization [27], as well as by polymer-analogous reactions during the interaction of functional groups of chitosan and preformed polymer [28,29]. The grafting of vinyl [30], acrylic [31], fluorinated [23], organosilicon [32] and other polymers can be used to impart hydrophobicity to chitosan. This paper proposes to use copolymers based on alkyl methacrylates (AlMA) and glycidyl methacrylate (GMA) as modifiers that are capable of forming covalent bonds owing to the interaction of epoxy anchor groups with complementary groups on the surface of chitosan materials, while alkyl methacrylate comonomers will provide hydrophobic and oleophilic properties at the interface.

The purpose of this work was to study the effect of the length of alkyl side groups in glycidyl methacrylate and alkyl methacrylate copolymers on the wettability and sorption properties of surface-modified chitosan aerogels relative to water and liquid hydrocarbons (e.g., oil, diesel fuel, and synthetic motor oil) as well as to evaluate the biodegradability of the chitosan materials after modification by synthetic polymers.

## 2. Materials and Methods

### 2.1. Materials

Chitosan (M_w_ = 200 kDa, degree of deacetylation 83%) was purchased from Bioprogress (Russia). Glacial acetic acid (p.a.), methanol (c.p.), ammonia (25% water solution, p.a.) methyl ethyl ketone (c.p.), and diethylamine (pure) were purchased from Vekton (Russia). Glutaric aldehyde (GA, 25% aqueous solution) was purchased from Acros Organics. Glycidyl methacrylate (GMA, 98%), hexyl methacrylate (HeMA, 97%), decyl methacrylate (DMA, 96%), lauryl methacrylate (LMA, 96%), stearyl methacrylate (SMA, 97%), and azobisisobutyronitrile (AIBN, 98%) were purchased from Aldrich. Tetradecyl methacrylate (TDMA, 96%,) was purchased from J&K Scientific, Beijing, China. Synthetic motor oil (viscosity grade 5W30) produced by Total Quartz, winter diesel fuel produced by Rosneft, light oil (ρ = 0.803 g/cm^3^), distilled water, and deionized water were used.

### 2.2. Preparation of the Chitosan Aerogel

A 3% aqueous solution of GA was added to a 1% solution of chitosan in a 1% aqueous solution of acetic acid (at a molar ratio of NH_2_:C=O of 1:1) and stirred with a magnetic stirrer at 1500 rpm for 30 min at room temperature. The resulting gel was degassed in an ultrasonic bath for 10 min, placed in polypropylene molds and left for 72 h at room temperature. Then, it was frozen at −30 °C for 18 h, followed by complete defrosting at 25 °C. The resulting hydrogel was converted from the salt form to the primary form with an aqueous-alcoholic solution of ammonia (1:1 vol.); further washing to a neutral pH was performed with distilled water. The reduced hydrogels were frozen at −30 °C and then dried at −82 ± 2 °C under reduced pressure in a 2.5-L FreeZone freeze dryer from Labconco Plus (Kansas City, MO, USA).

### 2.3. Preparation of the Chitosan Films

Chitosan films were formed from 2% chitosan solutions in a 2% aqueous solution of acetic acid according to the previously described procedure [33]. A dispersion of 2.6 g of chitosan in 98 g of water was stirred for 30 min to swelling, after which 2.1 g of glacial acetic acid was added dropwise (to obtain a 2% solution of chitosan in 2% acetic acid). The mixture was stirred for an additional 3 h. The chitosan solution was cast into Petri dishes and dried at 40 °C under reduced pressure. The resulting product was washed with an alcoholic alkali solution (1:1) for 1 h and then repeatedly with distilled water. The films were dried at 40 °C under reduced pressure until constant weight. The dried films had a thickness of 30–40 μm.

### 2.4. Synthesis of GMA and AlMA Copolymers

The synthesis of GMA and AlMA copolymers was performed by a free-radical mechanism similar to the previously described procedure [34]. The synthesis of copolymers was conducted in MEK for 24 h at 70 °C with a GMA:AlMA mole ratio of 2.3:1. The overall concentration of monomers was 1 mol L^−1^. AIBN was used as an initiating agent. The copolymer was precipitated in cold methanol and then dried under reduced pressure until constant weight.

### 2.5. Surface Modification of the Chitosan-Based Materials

The modification of chitosan aerogels was performed by immersing test samples in a poly(GMA-co-AlMA) solution in methyl ethyl ketone with a copolymer concentration of 0.01–1 wt% for 30 min. Afterwards, the samples were removed from the solution, placed on dry glass Petri dishes and heated at 20–160 °C for 60 min. Chitosan aerogels were washed from adsorbed copolymer in a Soxhlet apparatus for 8 h using methyl ethyl ketone as the extractant. The weight gain of the samples determined gravimetrically following the grafting of GMA and AlMA copolymers and washing was ~4 ± 1% (average for a series of samples of *n* = 30).

Similarly, the modification of chitosan films was performed by immersing samples in a modified solution of poly(GMA-co-AlMA) copolymers in methyl ethyl ketone with a concentration of 3 wt% for 60 min. Afterwards, the samples were removed from the solution and thermostated at 140 °C for 60 min. The samples were purified from unreacted copolymer in a Soxhlet apparatus using methyl ethyl ketone as the solvent.

### 2.6. Amination of the Free Surface Epoxy Groups of the Obtained Materials

Chitosan films and aerogels modified with poly(GMA-co-TDMA) copolymer were placed in 20 wt% solution of diethylamine in methyl ethyl ketone for 24 h at 70 °C. Then, the materials were washed with methyl ethyl ketone in a Soxhlet apparatus for 8 h.

### 2.7. Methods

The elemental composition of the copolymers based on glycidyl methacrylate and alkyl methacrylates, original and modified chitosan materials was studied using a CHNOS Vario EL Cube elemental analyzer (Germany) via the 2 mg 70 s method. The analysis time for one sample was 10 min, and the rates of consumption of He and O_2_ were 230 and 38 mL/min, respectively, with an oxygen supply time of 70 s. The temperatures of the oxidation and reduction columns were 1150 and 850 °C, respectively. 

The molecular weight characteristics of the polymers were determined via gel permeation chromatography using a Shimadzu instrument (Japan) with columns filled with polystyrene gel with a pore size of 10^5^ and 10^4^ Å, tetrahydrofuran eluent, at 40 °C. A differential refractometer was used as a detector. Chromatograms were processed using the LCsolution software. Narrow-disperse standards of poly(methyl methacrylate) (PMMA) were used for calibration.

The Fourier-transform infrared (FTIR) spectroscopic studies of original and modified chitosan materials were performed using an InfraLUM FT-08 apparatus (Russia) in the range of 400–4000 cm^−1^ at a resolution of 0.7 cm^−1^.

The surface elemental composition was determined by X-ray photoelectron spectroscopy (XPS) using an ultra-high vacuum Multiprobe RM complex (Omicron Nanotechnology GmbH, Taunusstein, Germany) operating at a residual pressure of ~7 × 10^−9^ mbar. Mg Kα (E = 1253.6 eV, ΔE = 0.75 eV) was used to excite the photoemission. The diameter of the collection area of the hemispherical energy analyzer was 3 mm. We used a constant transmittance function mode with a transmittance energy value of 50 eV. The sampling rate for the O 1s, C 1s, N 1s, and F 1s lines was 0.2 eV, and the accumulation time at each energy point was 1 s. The survey spectrum was recorded at a rate of 1 eV and an accumulation time of 0.5 s at each energy point.

The chitosan aerogel structure was studied by scanning electron microscopy (SEM) using a Versa 3D instrument (FEI, Hillsboro, OR, USA) in low vacuum mode at a water vapor pressure in the chamber of 10–80 Pa and an accelerating voltage of 15–20 kV.

The study of the structural features of the chitosan aerogel involved a transverse fracture of the sample in liquid nitrogen (after a 60-min exposure) by a sharp application of force on the sample edges. To exclude prolonged contact of the prepared sample with laboratory air, the morphological features and elemental composition of the fracture surface were studied immediately after performing the abovementioned procedure.

The porosity (P) of the samples was calculated using the following formula:(1)$$P=1-\frac{p_{app}}{p_{tr}},$$
where *p_app_* is the apparent density of the sample (g/cm^3^), and *p_tr_* is the true density of the sample (g/cm^3^).

We determined the apparent density using the cylindrical aerogel samples, the height and diameter of which were measured using a digital caliper. The true density was determined by helium pycnometry using a Micromeritics AccuPyc 1330 pycnometer (Norcross, GA, USA).

The contact angle was measured using an OCA 15EC apparatus from DataPhysics (Filderstadt, Germany) according to the procedure described in [35]. The contact angle measurements were performed by applying 5 µL drops of test liquid on the surface of the material at room temperature in several environments, i.e., air, nonpolar liquid (diesel fuel), and polar liquid (deionized water). The contact angle of a sessile drop was calculated using the Young–Laplace method.

The sorption capacity was evaluated by the mass variation of the samples after placing them in a container with the appropriate liquid according to ASTM F726-12. Prior to testing, chitosan aerogel samples were dried at 110 °C to constant weight (initial weight *m*_0_). A sample weighing 0.1 ± 0.01 g was immersed in a 100-mL weighing bottle filled with 50 mL of test liquid. After 15 min and 24 h of testing, the sample was removed from the container and placed on a wire rack for 30 s to drain off excess fluid. The sorption capacity C was calculated according to the following formula:(2)$$C=\frac{m_{i}-m_{0}}{m_{0}},$$
where *m_i_* is the sample weight after 15 min or 24 h after the start of the test and 30 s of draining off on the wire rack (g), and *m*_0_ is the initial weight of the sample (g).

The biodegradability study was performed using the original and modified samples of chitosan films in soil. The 30 × 30 mm samples with a thickness of 35 ± 5 μm were buried at least 5 cm deep in a container filled with soil at a pH of the aqueous extract of 6–7.5, humidity of 60–70%, and temperature of 23 ± 5 °C. Every 15 days, the samples were extracted from the soil, washed with distilled water, dried at 100 °C to constant weight, and weighed. The percent of weight loss was calculated using the following equation:(3)$$\mathrm{WL}=\frac{m_{i}-m_{0}}{m_{0}}\times 100,$$
where *m_i_* is the sample weight after contact with soil (g), and m_0_ is the initial weight of the sample (g).

## 3. Results and Discussion

### 3.1. Synthesis and Characterization of Chitosan Materials Modified by GMA and AlMA Copolymers

The surface of chitosan aerogels with GMA and AlMA copolymers was modified by impregnation according to the “grafting onto” approach [36]. The “grafting onto” approach involves the preliminary synthesis of polymers, the functional groups of which react with complementary functional groups on the modified surface. This modification approach preserves such matrix properties as the porous structure and the biodegradability of chitosan. Modification of the chitosan aerogel with GMA and AlMA copolymers using the abovementioned method will yield a surface layer representing chitosan macromolecular chains with randomly distributed graft copolymer branches formed as a result of the reaction between oxirane groups of GMA and hydroxy and amino groups of chitosan (Figure 1).

One advantage of using the “grafting onto” approach is the possibility of determining the molecular characteristics of the modifier polymers. The composition and molecular weight characteristics of poly(GMA-co-AlMA) are shown in Table 1.

Based on elemental analysis, the ratio of monomeric units in the resulting copolymers was found to be similar to the theoretical molar ratio. The copolymers are characterized by a narrow molecular weight distribution and low molecular weights, while the similar values of these indicators allow us to study the effect of the composition and structure of the polymer modifiers on the hydrophobic and sorption properties of the surface of chitosan materials. 

To confirm the grafting of GMA and AlMA copolymers on the surface of chitosan aerogels, the FTIR spectroscopic studies were performed using attenuated total reflection (Figure 2). The FTIR spectra of the modified samples have a band at 1728 cm^−1^, which corresponds to stretching vibrations of the carbonyl group in esters [37] and confirms the grafting of poly(GMA-co-AlMA) copolymers on the chitosan surface. Grafting of copolymers predominantly at the amino group was confirmed by a reduction in the intensity of the absorption band of NH_2_ groups at 1582 cm^−1^.

At the stage of aerogel formation, GA is used to create the three-dimensional structure of the material, and the mechanism of its interaction with chitosan is complex and not limited to mere interaction of aldehyde groups with amino groups [38]. Based on the elemental analysis of the carbon to nitrogen ratio (Table 2) for the chitosan film that did not contain GA and the chitosan aerogel crosslinked with GA, the content of the latter was 10.38%, which corresponds to the ratio of functional groups of NH_2_:C=O equal to 2.1:1 in the resulting chitosan aerogel. Consequently, crosslinking does not involve at least half of the chitosan amino groups, which can be involved in the reaction with epoxy groups of polymer modifiers. The attachment of GMA and AlMA copolymers on the surface of chitosan aerogels results in an increase in the C/N ratio in which the content of polymer modifiers was determined to be 2.5–3%, which is consistent with the gravimetric determination.

To identify free epoxy groups on the surface of chitosan aerogels modified with glycidyl methacrylate copolymers, they were aminated using diethylamine. The use of low-molecular-weight amines with small alkyl substituents provides the highest degree of conversion of epoxy groups in glycidyl methacrylate links owing to the leveling of steric hindrances [39,40,41]. In addition, to increase the availability of epoxy groups, amination was performed in a medium of methyl ethyl ketone, which is a good solvent for modifiers. The amination of chitosan aerogel modified with poly(GMA-co-TDMA) resulted in an increase in carbon and nitrogen content. The content of the reacted diethylamine determined from the elemental composition was 0.13%; the molar ratio of the secondary amino group:epoxy group was 1:4.3, which corresponds to the conversion of epoxy groups in the poly(GMA-co-TDMA) copolymer grafted to the chitosan surface of 23.3%. Thus, up to 76.7% of the epoxy groups contained in poly(GMA-co-TDMA) could be involved in the chitosan aerogel modification.

XPS was used to determine the elemental composition of the surface (Figure 3, Table 3) and the bonding configurations between the elements (Table 4). As shown in Figure 3A, there are four peaks in the photoelectron spectrum of the C 1s region of the original chitosan aerogel: C=O (288.3 eV), C–O (286.3 eV), C–N (286.0 eV), and C–C (284.7 eV) [42,43]. The photoelectron spectrum of the N 1s region also consists of four peaks: –RN^+^ (402.8 eV), –NH– (400.3 eV), N–C (399.2 eV), and –NH_2_ (398.5 eV) [43,44,45]. The N/C ratio of 0.08 determined by the XPS method was comparable to the findings of the elemental analysis of the bulk volume of material, which was 0.14. The grafting of poly(GMA-co-TDMA) decreased the nitrogen and oxygen content to 1.0% and 20.4%, respectively, while the carbon content increased to 78.6%. In addition, an increase in the C–C bond intensity was observed in the C 1s region (Figure 3B), which is a consequence of the modifier containing a long hydrocarbon substituent C_14_H_29_. As the XPS spectra of the analyzed sample were obtained at a depth of no more than 10 nm, the detection of nitrogen in the spectrum allows us to conclude that the thickness of the formed coating on the aerogel surface was even thinner [46]. Further treatment of the aerogel with diethylamine increased the nitrogen content to 2.3% and increased the C–C bond concentration to 64.9% in the C 1s photoelectron region (Figure 3C). Thus, the XPS method confirms the grafting of poly(GMA-co-TDMA) to the surface of chitosan aerogels and the subsequent attachment of diethylamine to the free epoxy groups of the modifier.

SEM images (Figure 4) show the morphology of chitosan aerogel fractures before and after modification. The resulting chitosan aerogels have high porosity and low density: pore sizes are 100–300 μm, and the pore walls are made of films with a thickness of 0.6–1 μm.

The modification of aerogels using the solutions based on GMA and AlMA copolymers does not alter the morphology and sizes of pores, which indicates the process of modification on the surface of the aerogel walls. The gravimetrically determined weight gain of the samples is only 4 ± 1%, which has almost no effect on the change in apparent density. In addition, there is no effect of modification on pycnometrically determined true density and porosity. The physical characteristics of the resulting chitosan aerogels are presented in Table 5.

The surface modification of porous materials requires determining the optimal modification conditions, such as the concentration of the modifying solution and temperature, which are necessary and sufficient to impart the best hydrophobic properties. The data on evaluating the effect of these parameters on the obtained values of initial contact angles and water absorption of aerogels are presented in Table 6 and Table 7. During the modification of chitosan aerogels with GMA and TDMA copolymer solutions, the required concentration of the modifier was determined to be 0.1 wt%, and the modification temperature was 140 °C. Variation of the abovementioned parameters yielded high values for the initial contact angles of up to 150° and higher. However, not all conditions can ensure the stability of the hydrophobic state of the surface, which is determined from water absorption results.

### 3.2. Wettability and Sorption Properties of Chitosan Aerogels Modified by GMA and AlMA Copolymers

One way to evaluate the effectiveness of chitosan aerogels for selective sorption of hydrocarbon fractions is to measure surface wetting angles (Table 8). The original chitosan aerogels are characterized by hydrophilicity and instantaneous absorption of a drop of deionized water in air. At the same time, modification of chitosan aerogels with GMA and AlMA copolymers imparts superhydrophobic properties to the surface with contact angles up to 157° (Figure 5a), which are due not only to the chemical composition at the interface, but also to the multilevel surface texture [47,48].

The grafting of copolymers onto the surface of chitosan aerogels provides a directional change in properties at the interface. Of note, organic liquids wet most surfaces in air; thus, the oleophilicity of chitosan aerogels was assessed in deionized water using diesel fuel as a test wetting fluid (and vice versa, water wetting in diesel fuel). Immersion of unmodified aerogels in deionized water demonstrates the absorption of the medium (complete water wetting) owing to the hydrophilic nature of chitosan, with the samples showing superoleophobic properties with contact angles over 160° in diesel fuel. This occurs owing to the formation of an aqueous film on the surface of the unmodified aerogel, which prevents the material from wetting with diesel fuel (Figure 5d). Similarly, unmodified chitosan aerogels are superhydrophobic in diesel fuel (Figure 5e). The aerogel samples modified with GMA and AlMA copolymers are superoleophilic both in the air and in an aqueous medium (Figure 5b,c), but retain superhydrophobic properties when immersed in a nonpolar medium. Thus, the modification of the surface of chitosan aerogels with GMA and AlMA copolymers provides the optimal set of lyophilic properties required for the selective sorption of hydrocarbons.

According to the data presented in Table 9, modification of chitosan aerogels with GMA and AlMA copolymers leads to a decrease in the water absorption index. However, the minimum water absorption of 1.8 g/g is achieved in the case of grafting of GMA and SMA copolymers with the largest alkyl substituent among the modifiers used, which correlates with the values of contact angles in air (Table 8). The HeMA and GMA copolymer with a short alkyl substituent provides water absorption of chitosan aerogels up to 4.6 g/g, thus demonstrating the lowest efficiency among the studied modifiers. DMA, LMA, and TDMA copolymers can be attributed to the group of average indicators, the water absorptions of which were 2.1, 2.2 and 2.1 g/g, respectively, with their water repellency being similar to that of SMA copolymers. Despite the similar values of the initial contact angles for TDMA and SMA, the alkyl substituent length affects the water adsorption during prolonged contact, which indicates a less stable superhydrophobic state of the surface modified with GMA and TDMA copolymers.

The alkyl substituent in the polymeric modifier structure imparts hydrophobic properties to the surface by reducing the free surface energy and shielding hydrophilic fragments in the chitosan aerogel structure. Absorption of water by the modified samples can to some extent be explained by condensation of water vapor in the aerogel pores. Of note, under these experimental conditions, water adsorption by unmodified samples occurs almost instantaneously with the formation of swollen hydrogel, causing drying of the samples with their subsequent shrinkage and loss of the porous structure. Based on our findings (Table 9), it can be concluded that the high sorption rate of the hydrocarbon phase was observed in all chitosan aerogel samples. Even after 15 min of exposure, the sorption capacity almost reached the maximum values; during further contact with the sorbent medium up to 24 h, there was a slight increase in the amount of sorbent liquid (by 1.1 g/g on average). Further exposure does not produce an increase in the amount of sorbed hydrocarbon. Unlike unmodified aerogels, modified aerogels are characterized by lower indices of sorption capacity for diesel fuel, light crude, and synthetic motor oil. The average deviation from the value of the sorption capacity of unmodified aerogels is 3.4 g/g, which may be due to a decrease in the free volume of the aerogel as a result of grafting of copolymers and possible partial overlapping of micropores by the grafted copolymer. Of note, the low density of the samples provides them with buoyancy, which is preserved even when the sorbent is saturated.

For comparison, Table 10 shows some known sorbents and their sorption characteristics in relation to petroleum and petroleum products. We observed that the resulting chitosan aerogels modified with GMA and AlMA copolymers are significantly superior to their known counterparts in terms of sorption properties.

### 3.3. Biodegradability of Chitosan Materials Modified by GMA and AlMA Copolymers

Imparting hydrophobic properties to chitosan material inhibits hydrolytic processes, which are the basis of biodegradability [57]. One of the principal issues for the performed modification was the preservation of the biodegradability of the resulting materials, which in this study was determined by degrading films in soil (Figure 6 and Figure 7). Unmodified chitosan film loses approximately 90% of its weight when exposed to soil for 4–5 months. It can be assumed that complete biodegradation occurs after soil exposure for 5–6 months. After 65 days from the beginning of the experiment, the unmodified material was characterized by a weight loss of ~30%, whereas the samples modified with GMA and AlMA copolymers during this period showed the onset of destructive processes, which were preceded by an induction period associated with hydration of the modifier attached to the surface. After soil exposure for 5 months, chitosan films modified with poly(GMA-co-DMA) and poly(GMA-co-LMA) showed weight losses of up to 75% and 50%, respectively, which confirmed the preservation of the biodegradability of these materials.

## 4. Conclusions

This study evaluated the effect of the structure of grafted copolymers based on alkyl methacrylates with C_6_–C_18_ hydrocarbon side groups and glycidyl methacrylate on the lyophilic and sorption properties of chitosan aerogels. The unmodified chitosan aerogel was characterized by complete wetting and water absorption of 56.4 g/g, while the modification resulted in superhydrophobic aerogels characterized by wetting angles up to 157°. The copolymers based on alkyl methacrylates and glycidyl methacrylate grafted onto the surface of chitosan aerogels prevented the capillary effect relative to water and allowed the reduction of the material’s water absorption by a factor of 30 down to 1.8 g/g when modified with a GMA and SMA copolymer with the side alkyl group containing 18 carbon atoms. The GMA and HeMA copolymer has the least effective water-repellent effect and allows the reduction of water absorption by a factor of 12 down to 4.6 g/g. The length of the alkyl substituent of 10–14 carbon atoms in alkyl methacrylate yields similar water repellent properties in the water absorption range of 2.1–2.2 g/g. The oleophilic nature of the polymeric modifiers contributes to the sorption capacity of aerogels up to 44 g/g relative to various types of liquid hydrocarbons.

The resulting sorption and hydrophobic characteristics of the modified chitosan aerogels determine the possibility of selective sorption of hydrocarbons from water surfaces, which in conjunction with the preservation of biodegradability opens up prospects for the use of these materials as effective and environmentally friendly petroleum and oil sorbents.

## Figures and Tables

**Figure 1 polymers-14-02711-f001:**
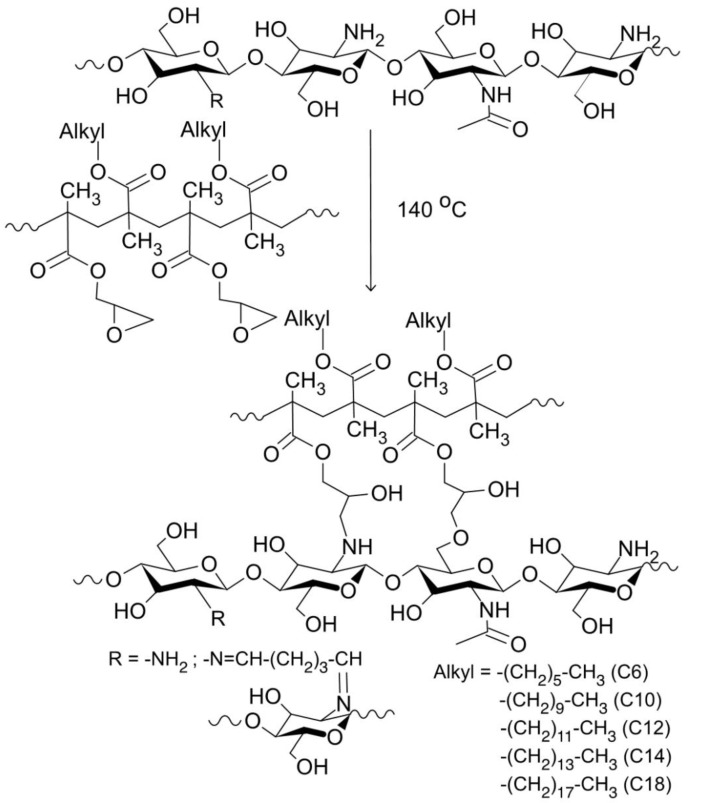
Attachment of GMA and AlMA copolymers on the surface of the chitosan aerogel.

**Figure 2 polymers-14-02711-f002:**
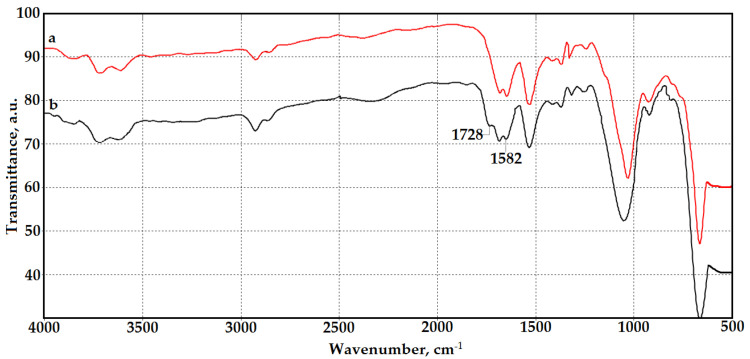
FTIR spectra of the original chitosan aerogel (**a**) and chitosan aerogel modified with poly(GMA-co-HeMA) copolymers (**b**).

**Figure 3 polymers-14-02711-f003:**
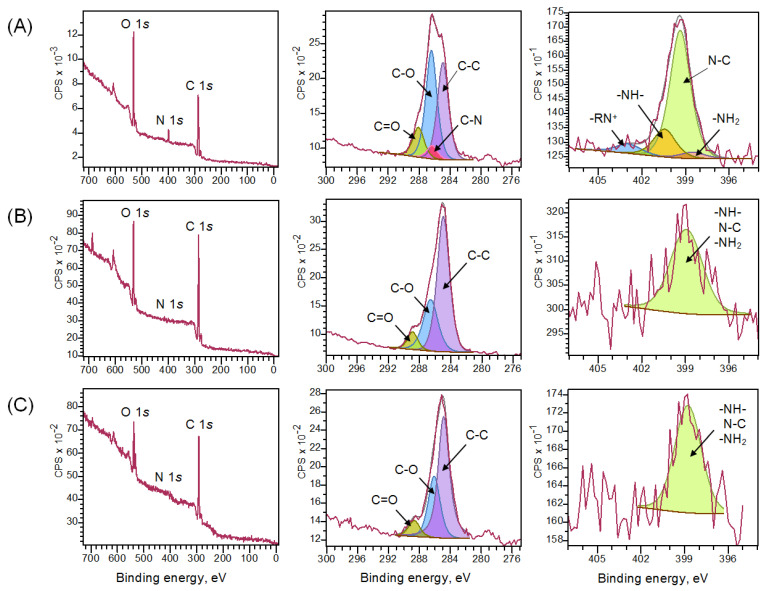
Survey scan, C 1s and N 1s core-level XPS spectra of the original chitosan aerogel (**A**), chitosan aerogel modified with poly(GMA-co-TDMA) copolymers (**B**), chitosan aerogel modified with poly(GMA-co-TDMA) copolymer and diethylamine (**C**).

**Figure 4 polymers-14-02711-f004:**
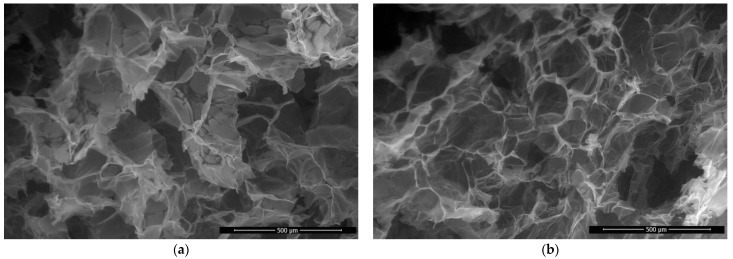
SEM images of fractures of the original chitosan aerogel (**a**) and chitosan aerogel modified with poly(GMA-co-HeMA) (**b**).

**Figure 5 polymers-14-02711-f005:**
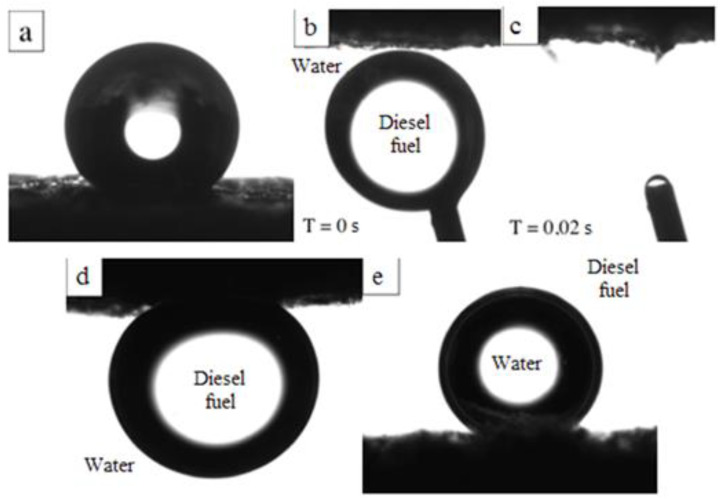
Images of test liquid drops in contact with the surface of chitosan aerogels: panel (**a**) shows a drop of deionized water in air medium on the surface of the chitosan aerogel modified with poly(GMA-co-DMA) copolymer; panels (**b**,**c**) show the process of diesel fuel drop absorption in deionized water medium by chitosan aerogel modified with poly(GMA-co-DMA) copolymer; panel (**d**) shows a drop of diesel fuel in deionized water medium on the surface of unmodified chitosan aerogel; and panel (**e**) shows a drop of deionized water in diesel fuel medium on the surface of unmodified chitosan aerogel.

**Figure 6 polymers-14-02711-f006:**
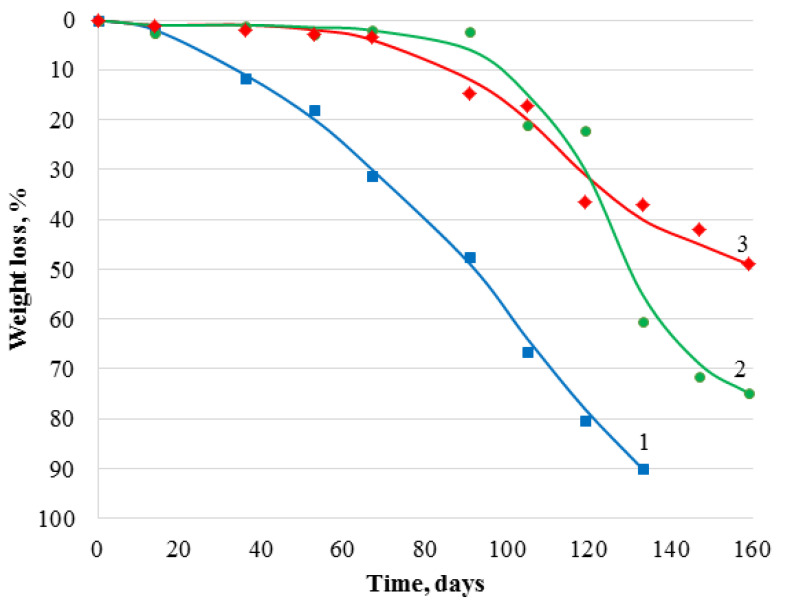
Weight loss of chitosan films under soil degradation conditions: 1-unmodified; 2-modified with poly(GMA-co-DMA); 3-modified with poly(GMA-co-LMA).

**Figure 7 polymers-14-02711-f007:**
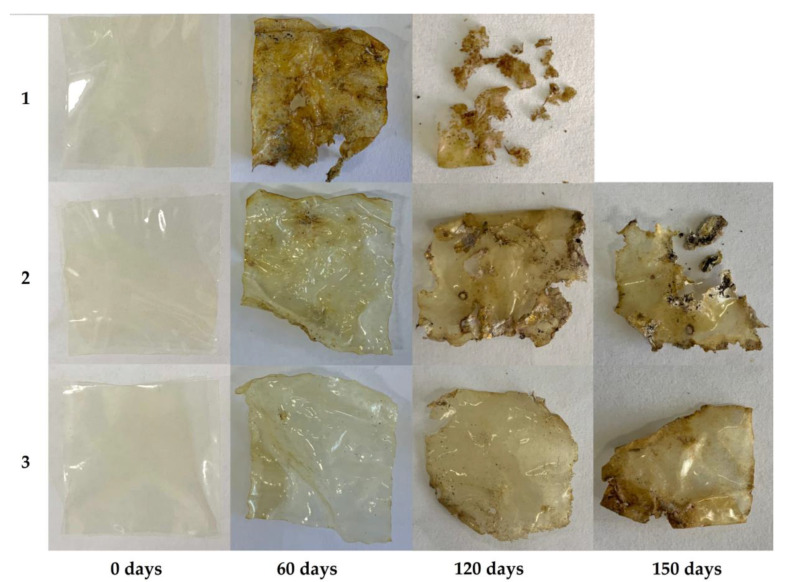
Chitosan films before and after soil degradation: 1-unmodified; 2-modified with poly(GMA-co-DMA); 3-modified with poly(GMA-co-LMA).

**Table 1 polymers-14-02711-t001:** Composition and molecular weight characteristics of GMA and AlMA copolymers.

Modifier	Molar Ratio		M_n_ × 10^−3^	M_w_ × 10^−3^	M_w_/M_n_
	Theoretical	Experimental			
Poly(GMA-co-HeMA)	2.3:1	2.0:1	39.8	77.8	1.9
Poly(GMA-co-DMA)		1.9:1	51.6	92.8	1.8
Poly(GMA-co-LMA)		2.2:1	71.1	159.5	2.2
Poly(GMA-co-TDMA)		1.9:1	44.6	81.8	1.8
Poly(GMA-co-SMA)		2.0:1	64.6	155.7	2.4

**Table 2 polymers-14-02711-t002:** Results of elemental analysis of chitosan materials before and after modification (by CHNOS method).

Chitosan Material	Elemental Composition, wt%			C/N	Modifier Content, wt%
	C	H	N		
Film	44.74	7.072	8.31	5.38	-
Aerogel	45.29	7.055	7.26	6.24	10.38 *
Aerogel, modified by poly(GMA-co-HeMA)	45.67	6.944	7.01	6.51	3.04
Aerogel, modified by poly(GMA-co-DMA)	45.76	6.919	7.07	6.47	2.49
Aerogel, modified by poly(GMA-co-TDMA)	45.51	6.895	7.02	6.48	2.50
Aerogel, modified by poly(GMA-co-TDMA) and diethylamine	45.61	6.733	7.05	6.47	0.13 **

* (GA content); ** (diethylamine content).

**Table 3 polymers-14-02711-t003:** Results of elemental analysis of the surface of chitosan aerogels before and after the modification obtained by the XPS method.

Chitosan Aerogel	Elemental Composition, at.%		
	O	N	C
Unmodified	28.2	5.8	66.0
With grafted poly(GMA-co-TDMA)	20.4	1.0	78.6
With grafted poly(GMA-co-TDMA) and diethylamine	19.3	2.3	78.4

**Table 4 polymers-14-02711-t004:** Concentrations of constituent photoelectron regions of chitosan aerogels before and after modification obtained by the XPS method.

Chitosan Aerogel	C 1*s*				N 1*s*			
	C–C	C–O	C=O	C–N	N–C	–RN^+^	–NH–	–NH_2_
Unmodified	34.9	43.8	12.0	9.2	72.3	6.4	16.9	4.3
With grafted poly(GMA-co-TDMA)	61.0	32.3	6.7	0.0	100.0	0.0	0.0	0.0
With grafted poly(GMA-co-TDMA) and diethylamine	64.9	26.3	8.8	0.0	100.0	0.0	0.0	0.0

**Table 5 polymers-14-02711-t005:** Physical characteristics of chitosan aerogels.

Chitosan Aerogel	Apparent Density, g/cm^3^	True Density, g/cm^3^	Porosity, %	Pore Diameter, µm	Pore Wall Thickness, µm
Unmodified	0.021 ± 0.003	1.357 ± 0.011	98.5 ± 0.4	100–300	0.6–1
With grafted poly(GMA-co-TDMA)	0.022 ± 0.003	1.364 ± 0.009	98.4 ± 0.4		

**Table 6 polymers-14-02711-t006:** Initial contact angle and water absorption of chitosan aerogels modified at 140 °C for 1 h using poly(GMA-co-TDMA) solutions at varying concentrations.

Concentration of the Modifier in Solution, wt%	Initial Contact Angle, °	Water Absorption (After 24 h), g/g
0.01	150 ± 7	26.4
0.02	151 ± 4	7.6
0.04	152 ± 3	5.5
0.08	155 ± 4	2.4
0.1	157 ± 2	2.1
0.2	157 ± 2	2.2
0.5	157 ± 2	2.2
1	157 ± 2	2.1

**Table 7 polymers-14-02711-t007:** Initial contact angle and water absorption of chitosan aerogels modified for 1 h using poly(GMA-co-TDMA) solutions with 0.1 wt% concentration at varying temperatures.

Modification Temperature, °C	Initial Contact Angle, °	Water Absorption (After 24 h), g/g
20	149 ± 5	18.4
40	145 ± 7	18.7
60	152 ± 5	18.1
80	154 ± 4	17.2
100	153 ± 4	3.5
120	154 ± 4	2.8
140	157 ± 2	2.1
150	157 ± 2	2.3
160	157 ± 2	2.2

**Table 8 polymers-14-02711-t008:** Wettability of chitosan aerogels as a result of modification with GMA and AlMA copolymers.

Chitosan Aerogel with Grafted GMA and AlMA Copolymers	Contact Angle in the “Wetting Agent/Medium” System, °		
	Deionized Water in air	Deionized Water in Diesel Fuel	Diesel Fuel in Deionized Water
unmodified	Wetted	160 ± 2	162 ± 2
Poly(GMA-co-HeMA)	146 ± 2	162 ± 2	Wetted
Poly(GMA-co-DMA)	152 ± 2		
Poly(GMA-co-LMA)	153 ± 3		
Poly(GMA-co-TDMA)	157 ± 2		
Poly(GMA-co-SMA)	157 ± 3		

**Table 9 polymers-14-02711-t009:** Sorption capacity of chitosan aerogels as a result of modification with GMA and AlMA copolymer solutions.

Chitosan Aerogel with Grafted GMA and AlMA Copolymers	Sorption Capacity, g/g							
	Distilled Water		Synthetic Motor Oil		Diesel Fuel		Light Oil	
	15 min	24 h	15 min	24 h	15 min	24 h	15 min	24 h
Unmodified	53.7	56.4	42.6	44.3	37.1	37.5	35.7	36.3
Poly(GMA-co-HeMA)	1.3	4.6	37.9	39.6	35.2	35.7	31.8	32.3
Poly(GMA-co-DMA)	1.0	2.1	41.4	43.5	32.8	33.5	30.9	32.3
Poly(GMA-co-LMA)	1.0	2.2	35.4	37.7	33.4	33.6	34.4	34.7
Poly(GMA-co-TDMA)	0.9	2.1	42.2	42.5	31.5	31.7	31.1	33.1
Poly(GMA-co-SMA)	0.8	1.8	43.0	44.0	34.3	34.6	34.3	34.9

**Table 10 polymers-14-02711-t010:** Sorption properties of some petroleum and oil sorbents.

Sorbent	Type of Sorbent Liquid	Sorption Capacity, g/g	Reference
Zeolite	Engine oil	0.4–0.9	[49]
Moss	Engine oil	28.4	[50]
Rice husk	Gasoline	3.7	[51]
	Diesel	5.5	
	Light crude oil	6.0	
	Motor oil	7.5	
	Heavy crude oil	9.2	
Butyl rubber	Toluene	17.8	[52]
	Gasoline	16.7	
	Diesel	20.3	
	Fuel oil	15.4	
	Crude oil	23.0	
	Olive oil	7.9	
Polypropylene fiber	Diesel	17.1	[53]
	High-density oil	18.8	
Cellulose aerogel	Crude oil	18.4–20.5	[54]
Silica aerogel	Diesel oil	9.6	[55]
Chitosan aerogel	Crude oil	41.1	[56]
	Diesel	31.1	
Chitosan aerogel	Synthetic motor oil	44.0	Current study
	Diesel fuel	35.7	
	Light oil	34.9	

## Data Availability

Not applicable.

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
