# Peer review of "Lyophilic and Sorption Properties of Chitosan Aerogels Modified with Copolymers Based on Glycidyl Methacrylate and Alkyl Methacrylates"

_polymers, 2022, doi:10.3390/polym14132711_

Round 1
Reviewer 1 Report
In this manuscript, the influence of the structure of copolymers on the wettability and sorption properties of chitosan aerogels was investigated in detail. The adsorption performance of the modified aerogels was improved dramatically, and the obtained aerogels retained the biodegradation ability. The manuscript was in good organizing and writing, therefore, a minor revision of this manuscript is recommended.
(1) The last paragraph of the “Introduction” is only one sentence, which should be divided into at least two sentence.
(2) Line 86, the present format of the density should be revised
(3) In the section of “2.4”, the synthesis process of copolymers should be provided.
(4) The IR spectra should present like the XPS spectra with a box.
(5) The physical characteristics of chitosan aerogels was shown in Table 5, however, the physical characteristics of modified chitosan aerogels was not provided. Though there is no obvious difference in the SEM image, while the modification is under high temperature and with a Soxhlet.
(6) The Figure 6 only shows the degradation behaviour of two modified materials, does other modified materials have the same degradation ability?
Author Response
Thanks to the reviewer for carefully reading the manuscript. We have taken into account the comments to improve the quality of the article.
We have revised our paper. Changes in the manuscript are marked by yellow color.
Reviewer 2 Report
This work looks similar work of the author's previous work with slightly different. Still, it seems to be worth publishing. Some of the major and minor issues need to be solved before final publications. Many inconsistencies throughout the manuscript. Grammar mistake. Please revise carefully before final publications.
Line 48, “film-forming capacity may be lost”, this phrases please double-check with references and insert another good references and correct it. Line 53, what is the meaning of “low-molecular-weight aldehydes: Please clarify it and rewrite it. Line 78, M or Mw? Check it.
In section 2.1, please make full sentences. Can authors use the abbreviation of chitosan as chitosan repeatedly using in the whole manuscript? Line 89, NH2 write superscript or subscript clearly, Check throughout the manuscript. Line 31-38, insert this current reference Materials Letters 316 (2022) 132046. Line -30 C. deg. The symbol is missing. Check carefully Line 96, 2oC make space between o and C throughout the manuscript. Section 2.3 please describe briefly the preparation method. Line 66 and 102, same name, first use short name then use throughout the manuscript. Check others also. Line 112 remove. dot mark in wt.%. Line 112-113 details needed. Line 24 hours please write as 24 h. Indicate the following references in the text accordingly https://doi.org/10.1016/j.jcou.2022.101958, Cellulose 29 (2022) 2399-2411, International Journal of Biological Macromolecules 195 (2022) 75-85International Journal of Biological macromolecules 136 (2019) 661-667, In table 10, put this work information.
Section 2.5 is not a clear determination of epoxy groups. Line 123, check O2. Line 128, check 40oC? Line 132-133, details expt. Condition needed for FTIR. Line 161, remove (–) in 5-µL. Divide in section-wise, results and discussion part.
In Biodegradability put the formula of weight loss. Figure 1 needs more details to mark the arrow mark for the reaction mechanism. Need solid-state NMR analysis. Without this, it is hard to prove that grafting.
Figure 2 please a and b instead of 1 and 2. The graph is unprofessional. Make it professional using scientific software. Show y-axis. What is Review in Fig 3 caption? What is table 5? Please provide other sample information. Authors must present soil degradation film digital photos. Check reference no 23.
Author Response

(The authors gave the same response as above.)

Round 2
Reviewer 2 Report
Accept! Next time please provide cover letter to each reviewers.